# WRPN: Wide Reduced-Precision Networks

**Asit Mishra, Eriko Nurvitadhi, Jeffrey J Cook & Debbie Marr**
Accelerator Architecture Lab
Intel Labs
{asit.k.mishra,eriko.nurvitadhi,jeffrey.j.cook,debbie.marr}@intel.com

## Abstract

For computer vision applications, prior works have shown the efficacy of reducing numeric precision of model parameters (network weights) in deep neural networks. Activation maps, however, occupy a large memory footprint during both the training and inference step when using mini-batches of inputs. One way to reduce this large memory footprint is to reduce the precision of activations. However, past works have shown that reducing the precision of activations hurts model accuracy. We study schemes to *train* networks from scratch using reduced-precision activations without hurting accuracy. We reduce the precision of activation maps (along with model parameters) and increase the number of filter maps in a layer, and find that this scheme matches or surpasses the accuracy of the baseline full-precision network. As a result, one can significantly improve the execution efficiency (e.g. reduce dynamic memory footprint, memory bandwidth and computational energy) and speed up the training and inference process with appropriate hardware support. We call our scheme WRPN - wide reduced-precision networks. We report results and show that WRPN scheme is better than previously reported accuracies on ILSVRC-12 dataset while being computationally less expensive compared to previously reported reduced-precision networks.

## 1 Introduction

A promising approach to lower the compute and memory requirements of convolutional deep-learning workloads is through the use of low numeric precision algorithms. Operating in lower precision mode reduces computation as well as data movement and storage requirements. Due to such efficiency benefits, there are many existing works which propose low-precision deep neural networks (DNNs) (Zhou et al., 2017; Lin et al., 2015; Miyashita et al., 2016; Gupta et al., 2015b; Vanhoucke et al., 2011), even down to 2-bit ternary mode (Zhu et al., 2016; Li & Liu, 2016; Venkatesh et al., 2016) and 1-bit binary mode (Zhou et al., 2016; Courbariaux & Bengio, 2016; Rastegari et al., 2016; Courbariaux et al., 2015; Umuroglu et al., 2016). However, majority of existing works in low-precision DNNs sacrifice accuracy over the baseline full-precision networks. Further, most prior works target reducing the precision of the model parameters (network weights). This primarily benefits the inference step only when batch sizes are small.

We observe that activation maps (neuron outputs) occupy more memory compared to the model parameters for batch sizes typical during training. This observation holds even during inference when batch size is around eight or more. Based on this observation, we study schemes for training and inference using low-precision DNNs where we reduce the precision of activation maps as well as the model parameters without sacrificing network accuracy.

To improve both execution efficiency and accuracy of low-precision networks, we reduce both the precision of activation maps and model parameters and increase the number of filter maps in a layer. We call networks using this scheme wide reduced-precision networks (WRPN) and find that this scheme compensates or surpasses the accuracy of the baseline full-precision network. Although the number of raw compute operations increases as we increase the number of filter maps in a layer, the compute bits required per operation is now a fraction of what is required when using full-precision operations (e.g. going from FP32 AlexNet to 4-bits precision and doubling the number of filters increases the number of compute operations by 4x, but each operation is 8x more efficient than FP32).

WRPN offers better accuracies, while being computationally less expensive compared to previously reported reduced-precision networks. We report results on AlexNet (Krizhevsky et al., 2012), batch-normalized Inception (Ioffe & Szegedy, 2015), and ResNet-34 (He et al., 2015) on ILSVRC-12 (Russakovsky et al., 2015) dataset. We find *4-bits to be sufficient for training* deep and wide models while achieving similar or better accuracy than baseline network. With 4-bit activation and 2-bit weights, we find the accuracy to be at-par with baseline full-precision. Making the networks wider and operating with 1-bit precision, we close the accuracy gap between previously reported binary networks and show state-of-the art results for ResNet-34 (69.85% top-1 with 2x wide) and AlexNet (48.04% top-1 with 1.3x wide). To the best of our knowledge, *our reported accuracies with binary networks and 4-bit precision are highest to date.*

Our reduced-precision quantization scheme is hardware friendly allowing for efficient hardware implementations. To this end, we evaluate efficiency benefits of low-precision operations (4-bits to 1-bits) on Titan X GPU, Arria-10 FPGA and ASIC. We see that FPGA and ASIC can deliver significant efficiency gain over FP32 operations (6.5x to 100x), while GPU cannot take advantage of very low-precision operations.

## 2 MOTIVATION FOR REDUCED-PRECISION ACTIVATION MAPS

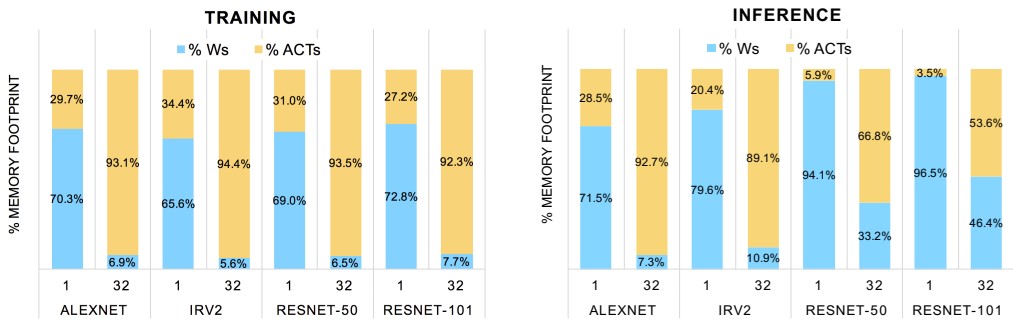

Figure 1: Memory footprint of activations (ACTs) and weights (W) during training and inference for mini-batch sizes 1 and 32.

While most prior works proposing reduced-precision networks work with low precision weights (e.g. work in Courbariaux & Bengio (2016); Zhu et al. (2016); Zhou et al. (2016); Venkatesh et al. (2016); Li & Liu (2016); Courbariaux et al. (2015); Umuroglu et al. (2016)), we find that activation maps occupy a larger memory footprint when using mini-batches of inputs. Using mini-batches of inputs is typical in training of DNNs and cloud-based batched inference (Jouppi et al., 2017). Figure 1 shows memory footprint of activation maps and filter maps as batch size changes for 4 different networks (AlexNet, Inception-Resnet-v2 (Szegedy et al., 2016), ResNet-50 and ResNet-101) during the training and inference steps.

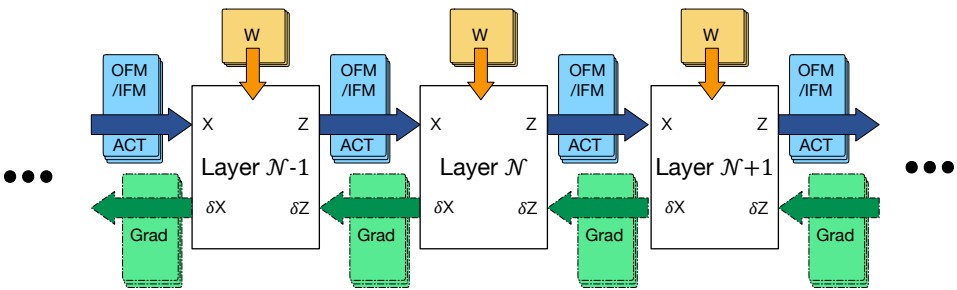

Figure 2: Memory requirements of a feed forward convolutional deep neural network. Orange boxes denote weights (W), blue boxes are activations (ACT) and green boxes are gradient-maps (Grad).

As batch-size increases, because of filter reuse across batches of inputs, activation maps occupy significantly larger fraction of memory compared to the filter weights. This aspect is illustrated in

Figure 2 which shows the memory requirements of a canonical feed-forward DNN for a hardware accelerator based system (e.g. GPU, FPGA, PCIe connected ASIC device, etc.). During training, the sum of all the activation maps (ACT) and weight tensors (W) are allocated in device memory for forward pass along with memory for gradient maps during backward propagation. The total memory requirements for training phase is the sum of memory required for the activation maps, weights and the maximum of input gradient maps ($\delta Z$) and maximum of back-propagated gradients ($\delta X$). During inference, memory is allocated for input (IFM) and output feature maps (OFM) required by a single layer, and these memory allocations are reused for other layers. The total memory allocation during inference is then the maximum of IFM and maximum of OFM required across all the layers plus the sum of all W-tensors. At batch sizes 128 and more, activations start to occupy more than 98% of total memory footprint during training.

Overall, reducing precision of activations and weights reduces memory footprint, bandwidth and storage while also simplifying the requirements for hardware to efficiently support these operations.

## 3 WRPN SCHEME AND STUDIES ON ALEXNET

Based on the observation that activations occupy more memory footprint compared to weights, we reduce the precision of activations to speed up training and inference steps as well as cut down on memory requirements. However, a straightforward reduction in precision of activation maps leads to significant reduction in model accuracy (Zhou et al., 2016; Rastegari et al., 2016).

We conduct a sensitivity study where we reduce precision of activation maps and model weights for AlexNet running ILSVRC-12 dataset and train the network from scratch. Table 1 reports our findings. Top-1 single-precision (32-bits weights and activations) accuracy is 57.2%. The accuracy with binary weights and activations is 44.2%. This is similar to what is reported in Rastegari et al. (2016). $32bA$ and $2bW$ data-point in this table is using Trained Ternary Quantization (TTQ) technique (Zhu et al., 2016). All other data points are collected using our quantization scheme (described later in Section 5), all the runs have same hyper-parameters and training is carried out for the same number of epochs as baseline network. To be consistent with results reported in prior works, we do not quantize weights and activations of the first and last layer.

We find that, in general, reducing the precision of activation maps and weights hurts model accuracy. Further, reducing precision of activations hurts model accuracy much more than reducing precision of the filter parameters. We find TTQ to be quite effective on AlexNet in that one can lower the precision of weights to 2b (while activations are still FP32) and not lose accuracy. However, we did not find this scheme to be effective for other networks like ResNet or Inception.

Table 1: AlexNet top-1 validation set accuracy % as precision of activations (A) and weight(W) changes. All results are with end-to-end training of the network from scratch. − is a data-point we did not experiment for.

|      | 32b A | 8b A | 4b A | 2b A | 1b A |
|------|-------|------|------|------|------|
| 32b W | 57.2 | 54.3 | 54.4 | 52.7 | – |
| 8b W | – | 54.5 | 53.2 | 51.5 | – |
| 4b W | – | 54.2 | 54.4 | 52.4 | – |
| 2b W | 57.5 | 50.2 | 50.5 | 51.3 | – |
| 1b W | 56.8 | – | – | – | 44.2 |

Table 2: AlexNet 2x-wide top-1 validation set accuracy % as precision of activations (A) and weights (W) changes.

|      | 32b A | 8b A | 4b A | 2b A | 1b A |
|------|-------|------|------|------|------|
| 32b W | 60.5 | 58.9 | 58.6 | 57.5 | 52.0 |
| 8b W | – | 59.0 | 58.8 | 57.1 | 50.8 |
| 4b W | – | 58.8 | 58.6 | 57.3 | – |
| 2b W | – | 57.6 | 57.2 | 55.8 | – |
| 1b W | – | – | – | – | 48.3 |

To re-gain the model accuracy while working with reduced-precision operands, we increase the number of filter maps in a layer. Although the number of raw compute operations increase with widening the filter maps in a layer, the bits required per compute operation is now a fraction of what is required when using full-precision operations. As a result, with appropriate hardware support, one can significantly reduce the dynamic memory requirements, memory bandwidth, computational energy and speed up the training and inference process.

Our widening of filter maps is inspired from Wide ResNet (Zagoruyko & Komodakis, 2016) work where the depth of the network is reduced and width of each layer is increased (the operand precision is still FP32). Wide ResNet requires a re-design of the network architecture. In our work, we maintain the depth parameter same as baseline network but widen the filter maps. We call our

approach WRPN - wide reduced-precision networks. In practice, we find this scheme to be very simple and effective - starting with a baseline network architecture, one can change the width of each filter map without changing any other network design parameter or hyper-parameters. Carefully reducing precision and simultaneously widening filters keeps the total compute cost of the network under or at-par with baseline cost.[1]

Table 2 reports the accuracy of AlexNet when we double the number of filter maps in a layer. With doubling of filter maps, AlexNet with 4-bits weights and 2-bits activations exhibits accuracy at-par with full-precision networks. Operating with 4-bits weights and 4-bits activations surpasses the baseline accuracy by 1.44%. With binary weights and activations we better the accuracy of XNOR-NET (Rastegari et al., 2016) by 4%.

When doubling the number of filter maps, AlexNet's raw compute operations grow by 3.9x compared to the baseline full-precision network, however by using reduced-precision operands the overall compute complexity is a fraction of the baseline. For example, with 4b operands for weights and activations and 2x the number of filters, reduced-precision AlexNet is just 49% of the total compute cost of the full-precision baseline (compute cost comparison is shown in Table 3).

Table 3: Compute cost of AlexNet 2x-wide vs. 1x-wide as precision of activations (A) and weights (W) changes.

|       | 32b A | 8b A | 4b A | 2b A | 1b A |
|-------|-------|------|------|------|------|
| 32b W | 3.9x  | 2.4x | 2.2x | 2.1x | 2.0x |
| 8b W  | 2.4x  | 1.0x | 0.7x | 0.6x | 0.6x |
| 4b W  | 2.2x  | 0.7x | 0.5x | 0.4x | 0.3x |
| 2b W  | 2.1x  | 0.6x | 0.4x | 0.2x | 0.2x |
| 1b W  | 2.0x  | 0.6x | 0.3x | 0.2x | 0.1x |

We also experiment with other widening factors. With 1.3x widening of filters and with 4-bits of activation precision one can go as low as 8-bits of weight precision while still being at-par with baseline accuracy. With 1.1x wide filters, at least 8-bits weight and 16-bits activation precision is required for accuracy to match baseline full-precision 1x wide accuracy. Further, as Table 3 shows, when widening filters by 2x, one needs to lower precision to at least 8-bits so that the total compute cost is not more than baseline compute cost. Thus, there is a trade-off between widening and reducing the precision of network parameters.

In our work, we trade-off higher number of raw compute operations with aggressively reducing the precision of the operands involved in these operations (activation maps and filter weights) while not sacrificing the model accuracy. Apart from other benefits of reduced precision activations as mentioned earlier, widening filter maps also improves the efficiency of underlying GEMM calls for convolution operations since compute accelerators are typically more efficient on a single kernel consisting of parallel computation on large data-structures as opposed to many small sized kernels (Zagoruyko & Komodakis, 2016).

## 4 STUDIES ON DEEPER NETWORKS

We study how our scheme applies to deeper networks. For this, we study ResNet-34 (He et al., 2015) and batch-normalized Inception (Ioffe & Szegedy, 2015) and find similar trends, particularly that 2-bits weight and 4-bits activations continue to provide at-par accuracy as baseline. We use TensorFlow (Abadi et al., 2015) and tensorpack for all our evaluations and use ILSVRC-12 train and val dataset for analysis.

### 4.1 RESNET

ResNet-34 has 3x3 filters in each of its modular layers with shortcut connections being 1x1. The filter bank width changes from 64 to 512 as depth increases. We use the pre-activation variant of ResNet and the baseline top-1 accuracy of our ResNet-34 implementation using single-precision

---

[1]Compute cost is the product of the number of FMA operations and the sum of width of the activation and weight operands.

32-bits data format is 73.59%. Binarizing weights and activations for all layers except the first and the last layer in this network gives top-1 accuracy of 60.5%. For binarizing ResNet we did not re-order any layer (as is done in XNOR-NET). We used the same hyper-parameters and learning rate schedule as the baseline network. As a reference, for ResNet-18, the gap between XNOR-NET (1b weights and activations) and full-precision network is 18% (Rastegari et al., 2016). It is also interesting to note that top-1 accuracy of single-precision AlexNet (57.20%) is lower than the top-1 accuracy of binarized ResNet-34 (60.5%).

Table 4: ResNet-34 top-1 validation accuracy % and compute cost as precision of activations (A) and weights (W) varies.

| Width | Precision | Top-1 Acc. % | Compute cost |
|---|---|---|---|
| 1x wide | 32b A, 32b W | 73.59 | 1x |
| | 1b A, 1b W | 60.54 | 0.03x |
| 2x wide | 4b A, 8b W | 74.48 | 0.74x |
| | 4b A, 4b W | 74.52 | 0.50x |
| | 4b A, 2b W | 73.58 | 0.39x |
| | 2b A, 4b W | 73.50 | 0.39x |
| | 2b A, 2b W | 73.32 | 0.27x |
| | 1b A, 1b W | 69.85 | 0.15x |
| 3x wide | 1b A, 1b W | 72.38 | 0.30x |

We experimented with doubling number of filters in each layer and reduce the precision of activations and weights. Table 4 shows the results of our analysis. Doubling the number of filters and 4-bits precision for both weights and activations beats the baseline accuracy by 0.9%. 4-bits activations and 2-bits (ternary) weights has top-1 accuracy at-par with baseline. Reducing precision to 2-bits for both weights and activations degrades accuracy by only 0.2% compared to baseline.

Binarizing the weights and activations with 2x wide filters has a top-1 accuracy of 69.85%. This is just 3.7% worse than baseline full-precision network while being only 15% of the cost of the baseline network. Widening the filters by 3x and binarizing the weights and activations reduces this gap to 1.2% while the 3x wide network is 30% the cost of the full-precision baseline network.

Although 4-bits precision seems to be enough for wide networks, we advocate for 4-bits activation precision and 2-bits weight precision. This is because with ternary weights one can get rid of the multipliers and use adders instead. Additionally, with this configuration there is no loss of accuracy. Further, if some accuracy degradation is tolerable, one can even go to binary circuits for efficient hardware implementation while saving 32x in bandwidth for each of weights and activations compared to full-precision networks. All these gains can be realized with simpler hardware implementation and lower compute cost compared to baseline networks.

To the best of our knowledge, our ResNet binary and ternary (with 2-bits or 4-bits activation) top-1 accuracies are state-of-the-art results in the literature including unpublished technical reports (with similar data augmentation (Mellempudi et al., 2017)).

## 4.2 BATCH-NORMALIZED INCEPTION

We applied WRPN scheme to batch-normalized Inception network (Ioffe & Szegedy, 2015). This network includes batch normalization of all layers and is a variant of GoogLeNet (Szegedy et al., 2014) where the 5x5 convolutional filters are replaced by two 3x3 convolutions with up to 128 wide filters. Table 5 shows the results of our analysis. Using 4-bits activations and 2-bits weight and doubling the number of filter banks in the network produces a model that is almost at-par in accuracy with the baseline single-precision network (0.02% loss in accuracy). Wide network with binary weights and activations is within 6.6% of the full-precision baseline network.

## 5 HARDWARE FRIENDLY QUANTIZATION SCHEME

We adopt the straight-through estimator (STE) approach in our work (Bengio et al., 2013). When quantizing a real number to $k$-bits, the ordinality of the set of quantized numbers is $2^k$. Mathemat-

Table 5: Batch-normalized Inception top-1 validation accuracy % and compute cost as precision of activations (A) and weights (W) varies.

| Width | Precision | Top-1 Acc. % | Compute cost |
|-------|-----------|--------------|--------------|
| 1x wide | 32b A, 32b W | 71.64 | 1x |
| 2x wide | 4b A, 4b W | 71.63 | 0.50x |
| | 4b A, 2b W | 71.61 | 0.38x |
| | 2b A, 2b W | 70.75 | 0.25x |
| | 1b A, 1b W | 65.02 | 0.13x |

ically, this small and finite set would have zero gradients with respect to its inputs. STE method circumvents this problem by defining an operator that has arbitrary forward and backward operations.

Prior works using the STE approach define operators that quantize the weights based on the expectation of the weight tensors. For instance, Ternary Weight Networks (TWN) (Li & Liu, 2016) uses a threshold and a scaling factor for each layer to quantize weights to ternary domain. In TTQ (Zhu et al., 2016), the scaling factors are learned parameters. XNOR-NET binarizes the weight tensor by computing the sign of the tensor values and then scaling by the mean of the absolute value of *each* output channel of weights. DoReFa uses a single scaling factor across the entire layer. For quantizing weights to $k$-bits, where $k > 1$, DoReFa uses:

$$w^k = 2 * quantize_k(\frac{tanh(w_i)}{2 * max(|\ tanh(w_i)\ |)} + \frac{1}{2}) - 1)$$

(1)

Here $w_k$ is the k-bit quantized version of inputs $w_i$ and $quantize_k$ is a quantization function that quantizes a floating-point number $w_i$ in the range $[0, 1]$ to a $k$-bit number in the same range. The transcendental $tanh$ operation constrains the weight value to lie in between $-1$ and $+1$. The affine transformation post quantization brings the range to $[-1, 1]$.

We build on these approaches and propose a much simpler scheme. For quantizing weight tensors we first hard constrain the values to lie within the range $[-1, 1]$ using min-max operation (e.g. tf.clip_by_val when using Tensorflow (Abadi et al., 2015)). For quantizing activation tensor values, we constrain the values to lie within the range $[0, 1]$. This step is followed by a quantization step where a real number is quantized into a $k$-bit number. This is given as, for $k > 1$:

$$w_k = \frac{1}{2^{k-1} - 1} round((2^{k-1} - 1) * w_i) \qquad \text{and} \qquad a_k = \frac{1}{2^k - 1} round((2^k - 1) * a_i)$$

(2)

Here $w_i$ and $a_i$ are input real-valued weights and activation tensor and $w_k$ and $a_k$ are their quantized versions. One bit is reserved for sign-bit in case of weight values, hence the use of $2^{k-1}$ for these quantized values. Thus, weights can be stored and interpreted using signed data-types and activations using un-signed data-types. With appropriate affine transformations, the convolution operations (the bulk of the compute operations in the network during forward pass) can be done using quantized values (integer operations in hardware) followed by scaling with floating-point constants (this scaling operation can be done in parallel with the convolution operation in hardware). When $k = 1$, for binary weights we use the Binary Weighted Networks (BWN) approach (Courbariaux et al., 2015) where the binarized weight value is computed based on the sign of input value followed by scaling with the mean of absolute values. For binarized activations we use the formulation in Eq. 2. We do not quantize the gradients and maintain the weights in reduced precision format.

For convolution operation when using WRPN, the forward pass during training (and the inference step) involves matrix multiplication of $k$-bits signed and $k$-bits unsigned operands. Since gradient values are in 32-bits floating-point format, the backward pass involves a matrix multiplication operation using 32-bits and $k$-bits operand for gradient and weight update.

When $k > 1$, the hard clipping of tensors to a range maps efficiently to min-max comparator units in hardware as opposed to using transcendental operations which are long latency operations. TTQ

and DoRefa (Zhou et al., 2016) schemes involve division operation and computing a maximum value in the input tensor. Floating-point division operation is expensive in hardware and computing the maximum in a tensor is an $O(n)$ operation. Additionally, our quantization parameters are static and do not require any learning or involve back-propagation like TTQ approach. We avoid each of these costly operations and propose a simpler quantization scheme (clipping followed by rounding).

## 5.1 EFFICIENCY IMPROVEMENTS OF REDUCED-PRECISION OPERATIONS ON GPU, FPGA AND ASIC

In practice, the effective performance and energy efficiency one could achieve on a low-precision compute operation highly depends on the hardware that runs these operations. We study the efficiency of low-precision operations on various hardware targets GPU, FPGA, and ASIC.

For GPU, we evaluate WRPN on Nvidia Titan X Pascal and for FPGA we use Intel Arria-10. We collect performance numbers from both previously reported analysis (Nurvitadhi et al., 2017) as well as our own experiments. For FPGA, we implement a DNN accelerator architecture shown in Figure 3(a). This is a prototypical accelerator design used in various works (e.g., on FPGA (Nurvitadhi et al., 2017) and ASIC such as TPU (Jouppi et al., 2017)). The core of the accelerator consists of a systolic array of processing elements (PEs) to perform matrix and vector operations, along with on-chip buffers, as well as off-chip memory management unit. The PEs can be configured to support different precision – (FP32, FP32), (INT4, INT4), (INT4, TER2), and (BIN1, BIN1). The (INT4, TER2) PE operates on ternary (+1,0,-1) values and is optimized to include only an adder since there is no need for a multiplier in this case. The binary (BIN1, BIN1) PE is implemented using XNOR and bitcount. Our RTL design targets Arria-10 1150 FPGA. For our ASIC study, we synthesize the PE design using Intel 14 nm process technology to obtain area and energy estimates.

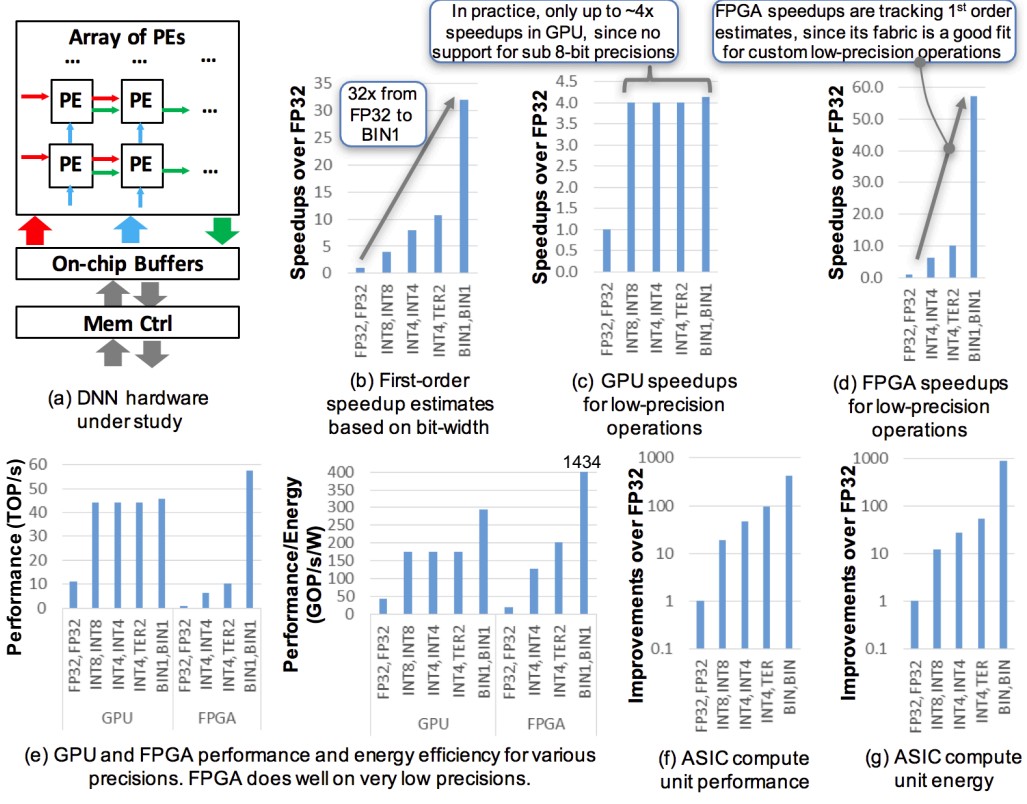

Figure 3: Efficiency improvements from low-precision operations on GPU, FPGA and ASIC.

Figure 3(b) - (g) summarize our analysis. Figure 3(b) shows the efficiency improvements using first-order estimates where the efficiency is computed based on number of bits used in the operation.

With this method we would expect (INT4, INT4) and (BIN1, BIN1) to be 8x and 32x more efficient, respectively, than (FP32, FP32). However, in practice the efficiency gains from reducing precision depend on whether the underlying hardware can take advantage of such low-precisions.

Figure 3(c) shows performance improvement on Titan X GPU for various low-precision operations relative to FP32. In this case, GPU can only achieve up to ∼4x improvements in performance over FP32 baseline. This is because GPU only provides first-class support for INT8 operations, and is not able to take advantage of the lower INT4, TER2, and BIN1 precisions. On the contrary, FPGA can take advantage of such low precisions, since they are amenable for implementations on the FPGAs reconfigurable fabric.

Figure 3(d) shows that the performance improvements from (INT4, INT4), (INT4, TER2), and (BIN1, BIN1) track well with the first-order estimates from Figure 3(b). In fact, for (BIN1, BIN1), FPGA improvements exceed the first-order estimate. Reducing the precision simplifies the design of compute units and lower buffering requirements on FPGA board. Compute-precision reduction leads to significant improvement in throughput due to smaller hardware designs (allowing more parallelism) and shorter circuit delay (allowing higher frequency). Figure 3(e) shows the performance and performance/Watt of the reduced-precision operations on GPU and FPGA. FPGA performs quite well on very low precision operations. In terms of performance/watt, FPGA does better than GPU on (INT4, INT4) and lower precisions.

ASIC allows for a truly customized hardware implementation. Our ASIC study provides insights to the upper bound of the efficiency benefits possible from low-precision operations. Figure 3(f) and 3(g) show improvement in performance and energy efficiency of the various low-precision ASIC PEs relative to baseline FP32 PE. As the figures show, going to lower precision offers 2 to 3 orders of magnitude efficiency improvements.

In summary, FPGA and ASIC are well suited for our WRPN approach. At 2x wide, our WRPN approach requires 4x more total operations than the original network. However, for INT4 or lower precision, each operation is 6.5x or better in efficiency than FP32 for FPGA and ASIC. Hence, WRPN delivers an overall efficiency win.

## 6 RELATED WORK

Reduced-precision DNNs is an active research area. Reducing precision of weights for efficient inference pipeline has been very well studied. Works like Binary connect (BC) (Courbariaux et al., 2015), Ternary-weight networks (TWN) (Li & Liu, 2016), fine-grained ternary quantization (Mellempudi et al., 2017) and INQ (Zhou et al., 2017) target precision reduction of network weights *while still using full-precision activations*. Accuracy is almost always degraded when quantizing the weights. For AlexNet on Imagenet, TWN loses 5% top-1 accuracy. Schemes like INQ, Sung et al. (2015) and Mellempudi et al. (2017) do fine-tuning to quantize the network weights and do not sacrifice accuracy as much but are not applicable for training networks from scratch. INQ shows promising results with 5-bits of precision.

XNOR-NET (Rastegari et al., 2016), BNN (Courbariaux & Bengio, 2016), DoReFa (Zhou et al., 2016) and TTQ (Zhu et al., 2016) target training as well. While TTQ targets weight quantization only, most works targeting activation quantization hurt accuracy. XNOR-NET approach reduces top-1 accuracy by 12% and DoReFa by 8% when quantizing both weights and activations to 1-bit (for AlexNet on ImageNet). Further, XNOR-NET requires re-ordering of layers for its scheme to work. Recent work in Graham (2017) targets low-precision activations and reports accuracy within 1% of baseline with 5-bits precision and logarithmic (with base $\sqrt{2}$) quantization. With fine-tuning this gap can be narrowed to be within 0.6% but not all layers are quantized.

Non-multiples of two for operand values introduces hardware inefficiency in that memory accesses are no longer DRAM or cache-boundary aligned and end-to-end run-time performance aspect is unclear when using complicated quantization schemes. We target *end-to-end training and inference*, using very simple quantization method and aim for *reducing precision without any loss in accuracy*. To the best of our knowledge, our work is the first to study reduced-precision deep and wide networks, and show accuracy at-par with baseline for as low a precision as 4-bits activations and 2-bits weights. *We report state of the art accuracy for wide binarized AlexNet and ResNet* while still being lower in compute cost.

Work by Gupta et al. (2015a) advocates for low precision fixed-point numbers for training. They show 16-bits to be sufficient for training on CIFAR10 dataset and find stochastic rounding to be necessary for training convergence. In our work here we focus on sub-8b training and like DoReFa scheme do not see stochastic rounding necessary when using full-precision gradients. Work by Seide et al. (2014) quantizes gradients before communication in a distributed computing setting. They use full precision gradients during the backward pass and quantize the gradients before sending them to other computation nodes (decreasing the amount of communication traffic over an interconnection network). For distributed training, we can potentially use this approach for communicating gradients across nodes.

## 7 CONCLUSIONS

We present the Wide Reduced-Precision Networks (WRPN) scheme for DNNs. In this scheme, the numeric precision of both weights and activations are significantly reduced without loss of network accuracy. This result is in contrast to many previous works that find reduced-precision activations to detrimentally impact accuracy; specifically, we find that 2-bit weights and 4-bit activations are sufficient to match baseline accuracy across many networks including AlexNet, ResNet-34 and batch-normalized Inception. We achieve this result with a new quantization scheme and by increasing the number of filter maps in each reduced-precision layer to compensate for the loss of information capacity induced by reducing the precision. We believe ours to be the first work to study the interplay between layer width and precision – with widening, the number of neurons in a layer increase; yet with reduced precision, we control overfitting and regularization.

We motivate this work with our observation that full-precision activations contribute significantly more to the memory footprint than full-precision weight parameters when using mini-batch sizes common during training and cloud-based inference; furthermore, by reducing the precision of both activations and weights the compute complexity is greatly reduced (40% of baseline for 2-bit weights and 4-bit activations).

The WRPN quantization scheme and computation on low precision activations and weights is hardware friendly making it viable for deeply-embedded system deployments as well as in cloud-based training and inference servers with compute fabrics for low-precision. We compare Titan X GPU, Arria-10 FPGA and ASIC implementations using WRPN and show our scheme increases performance and energy-efficiency for iso-accuracy across each. Overall, reducing the precision allows custom-designed compute units and lower buffering requirements to provide significant improvement in throughput.

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
