# OpenReview forum: "WRPN: Wide Reduced-Precision Networks"
_ICLR.cc/2018/Conference — Accept (Poster)_

### Official Review · AnonReviewer2 · 2017-11-27
**WRPN: Wide Reduced-Precision Networks**

**Rating:** 5
**Confidence:** 3

**Review:**

The paper studies the effect of reduced precision weights and activations on the performance, memory and computation cost of deep networks and proposes a quantization scheme and wide filters to offset the accuracy lost due to the reduced precision. The study is performed on AlexNet, ResNet and Inception on the Imagenet datasets and results show that accuracy matching the full precision baselines can be obtained by widening the filters on the networks.

Positives
- Using lower precision activations to save memory and compute seems new and widening the filter sizes seems to recover the accuracy lost due to the lower precision.

Negatives
- While the exhaustive analysis is extremely useful the overall technical contribution of the paper that of widening the networks is fairly small.
- The paper motivates the need for reduced precision weights from the perspective of saving memory footprint when using large batches. However, the results are more focused on compute cost. Also large batches are used mainly during training where memory is generally not a huge issue. Memory critical situations such as inference on mobile phones can be largely mitigated by using smaller batch sizes. It might help to emphasize the speed-up in compute more in the contributions.

---

> ### Author Response · Authors · 2017-12-04
> **Response to comments**
>
> Thank you for the comments and reviews. They are useful to us.
>
> Please see our response to AnonReviewer3 on the novelty aspect of our paper. Overall we believe ours is a simple technique that works and that is easier for programmers to adopt.
>
> We will clearly articulate the speed-up in compute for the final version of the paper. Specializing the hardware (e.g. by adding compute components that implement 2-bits, 4-bits, binary, 8bits, etc.)  would definitely speed-up inference times. Our ASIC and FPGA evaluations (Section-5.1) are an attempt to highlight this aspect. Current hardware platforms are not optimized for 2-bits and 4-bits. One other aspect of lowering memory footprint is that the working set size of the workload starts to fit on chip and by lowering accesses to DRAM memory, the compute core starts to see better performance and energy savings (DRAM accesses are expensive in latency and energy).

---

### Official Review · AnonReviewer1 · 2017-11-29
**High accuracy at low-precision**

**Rating:** 9
**Confidence:** 4

**Review:**

This is a well-written paper with good comparisons to a number of earlier approaches. It focuses on an approach to get similar accuracy at lower precision, in addition to cutting down the compute costs. Results with 2-bit activations and 4-bit weights seem to match baseline accuracy across the models listed in the paper.

Originality
This seems to be first paper that consistently matches baseline results below int-8 accuracy, and shows a promising future direction.

Significance
Going down to below 8-bits and potentially all the way down to binary (1-bit weights and activations) is a promising direction for future hardware design. It has the potential to give good results at lower compute and more significantly in providing a lower power option, which is the biggest constraint for higher compute today.

Pros:
- Positive results with low precision (4-bit, 2-bit and even 1-bit)
- Moving the state of the art in low precision forward
- Strong potential impact, especially on constrained power environments (but not limited to them)
- Uses same hyperparameters as original training, making the process of using this much simpler.

Cons/Questions
- They mention not quantizing the first and last layer of every network. How much does that impact the overall compute?
- Is there a certain width where 1-bit activation and weights would match the accuracy of the baseline model? This could be interesting for low power case, even if the "effective compute" is larger than the baseline.

---

> ### Author Response · Authors · 2017-12-04
> **Cons/Questions response**
>
> Thank you for the comments and review.
>
> Effect on compute of not quantizing first layer and last:
> The total number of FMA operations in first and last layer is ~3% for ResNet-34 (and 1.5% for ResNet-50). So the effect on overall compute is smaller for these layers if not negligible. In our work, the first layer and last layer's weights and activations are not quantized and neither are these layers' width increased.
>
> For the first and last layer, we find, we can quantize the weights to 8-bits (at most) without much loss in accuracy compared to keeping them at full-precision (~0.2% additional accuracy loss) while quantizing the other layers to 4bits activations and 2-bits weight. So, in theory we can use integer compute for these layers if not 2-bits and 4-bits precision to speed up compute.
>
> The primary reason we did not quantize the first and last layer is because - we wanted to fairly compare against prior proposals - the works we compared against in the paper do not quantize these layers.
>
> At what widening factor does 1-bit come at-par with baseline full-precision?
> Our very preliminary results tell us that this could probably happen at 3.5x-4x widening.
> We run into experimental evaluation issues when doing these experiments -- making the layers wider blows up the device memory requirements (since we "emulate" the binary and other low-precision knobs with FP32 precision in GPUs). We are working on performing these experiments with distributed TensorFlow set-up. The other aspect is to lower the batch-size and still use a single node set-up but we have to change the learning rates then.

---

> > ### Comment · AnonReviewer1 · 2017-12-04
> > **Comparison with MobileNets**
> >
> > Thanks for the responses!
> >
> > One question that comes up in most compression situations is whether the technique works with smaller networks/architectures e.g. MobileNets (https://research.googleblog.com/2017/06/mobilenets-open-source-models-for.html) that have already been somewhat optimized for mobile like targets. Since MobileNets also trade-off compute vs accuracy with a focus on compute can they still be compressed past 8-bits?

---

> > > ### Author Response · Authors · 2017-12-05
> > > **WRPN + MobileNets**
> > >
> > > We have an implementation (which is somewhat, slightly, crappy) of MobileNet that reaches Top-1 accuracy of 64.87% with .46 Giga-FLOPs (compared to 67.4% Top-1 accuracy which the official model reaches: https://github.com/tensorflow/models/blob/master/research/slim/nets/mobilenet_v1.md). We dont know the reason for this accuracy discrepancy and are debugging our implementation; because of this accuracy discrepancy we did not report WRPN results on MobileNet.
> > >
> > > Nevertheless,  this model with 2-bits weight and 4-bits activation gives Top-1 accuracy of 54.24% (i.e. a loss of 10.6%). With 2-bits weight and 4-bits activation and making the model 2x-wider, we get 64.66% Top-1 accuracy (a loss of 0.21% from our baseline). This also required the first layer to be widened. We think increasing by 2.1x or so should completely recover the accuracy.
> > > So, we think WRPN works for smaller networks.
> > >
> > > However, the more compact the model the harder it is to quantize to very low precision -- i.e. the accuracy loss is big with quantization to ternary or 4-bits precision. One interesting study we found is https://arxiv.org/pdf/1710.01878.pdf which shows a large sparse model is better than a small compact model. In the same theme, we believe a (somewhat) large quantized model is better than a small full-precision model.

---

> > > > ### Author Response · Authors · 2017-12-05
> > > > **Contd.**
> > > >
> > > > This is a good hardware-software co-design problem. 2-bit operands require simple hardware circuitry whereas int8, fp16 and fp32 require hardware multipliers. So, with wider or narrower models there should be a pareto curve for accuracy vs. model size vs. precision vs. hardware efficiency (performance, area, power) of the compute units at the precision used.

---

### Official Review · AnonReviewer3 · 2017-12-01
**Review of WRPN**

**Rating:** 5
**Confidence:** 4

**Review:**

This paper presents an simple and interesting idea to improve the performance for neural nets. The idea is we can reduce the precision for activations and increase the number of filters, and is able to achieve better memory usage (reduced). The paper is aiming to solve a practical problem, and has done some solid research work to validate that.  In particular, this paper has also presented a indepth study on AlexNet with very comprehensive results and has validated the usefulness of this approach.

In addition, in their experiments, they have demonstrated pretty solid experimental results, on AlexNet and even deeper nets such as the state of the art Resnet. The results are convincing to me.

On the other side, the idea of this paper does not seem extremely interesting to me, especially many decisions are quite natural to me, and it looks more like a very empirical practical study. So the novelty is limited.

So overall given limited novelty but the paper presents useful results, I would recommend borderline leaning towards reject.

---

> ### Author Response · Authors · 2017-12-04
> **Simple idea that (always) works**
>
> Thank you for the comments. We defend the novelty aspect of this paper in our response below.
>
> Novelty:
> Our paper targets quantization at no accuracy loss. We target network training on reduced precision hardware (H/W) considering a system-wide approach - system and on-chip memory footprint of activations is much more than weights.
> For cloud-based inference deployments (where large batch-size is typical) and during training, reducing precision of activations speeds up end-to-end runtime much more than reducing the precision of weights (Fig. 1).
> However, as our paper shows, reducing activation precision hurts accuracy much more than reducing weight precision. No prior work targets this aspect.
>
> Most prior works on reduced precision DNNs sacrifice accuracy and many prior works target reducing precision of just the weights. We show that there is no tradeoff in reducing precision - even during training - one can get the same accuracy as baseline by making the networks wider (yes, more raw compute operations, but still the compute cost is lower than baseline).
>
>
> 1. We believe lowering precision is one aspect (which is widely studied in literature) but it is important to lower precision without any loss in accuracy - no prior work has shown reduced-precision network (4-bits, 2-bits) training and inference without sacrificing accuracy.
> Also, our results with binary networks are state-of-the art and close the gap significantly between binary and 32b precision (e.g. less than 1.2% for ResNet-34).
>
>
> 2. We believe widening networks is a simple technique (which works) that is easy for programmers to experiment with for recovering accuracy with reduced precision. With WRPN: (a) model-size is smaller and, (b) run time and energy for end-to-end inference as well as training is lower than 32b networks.
>
> With widening, the number of neurons in a layer increase. Yet with reduced precision, we control overfitting and regularization. We believe, this aspect has not been studied before.

---

### Public Comment · (anonymous) · 2017-12-13
**XNOR is not used in Binary Net**

In the paper, the authors first hard constrain the values to lie within the range [−1, 1]; then quantizing activation tensor values and constrain the values to lie within the range [0, 1]. So after binarization, there are [-1, 0, 1] for weights and activations together which amounts more than 1-bit. This is different from the papers in the following where both weights and activations are constrained to +1 or 1 and potential XNOR+popcount implementation is promising.

[1] XNOR-Net: ImageNet Classification Using Binary Convolutional Neural Networks
[2] Binarized Neural Networks: Training Neural Networks with Weights and Activations Constrained to +1 or  1

---

> ### Author Response · Authors · 2017-12-13
> **We use {-,+} for weights and {0,1} for activations in binary mode**
>
> On page 6 our paper says (referring to Equation 2 for k), "When k = 1, for binary weights we use the Binary Weighted Networks (BWN) approach (Courbariaux et al., 2015) where the binarized weight value is computed based on the sign of input value followed by scaling with the mean of absolute values. For binarized activations we use the formulation in Eq. 2."
>
> In terms of TensorFlow code, this is implemented as:
>       m = tf.reduce_mean(tf.abs(x))
>       weights = tf.sign(x) * m
>
> The hard-clipping and quantization you mentioned in the comment is used for other values of k.
>
> If the text is not clear in the paper, let us know and we will fix it for the final version.

---

### Decision · Program_Chairs · 2018-01-29
**ICLR 2018 Conference Acceptance Decision**

**Decision:**

Accept (Poster)

**Comment:**

This paper explores the training of CNNs which have reduced-precision activations. By widening layers, it shows less of an accuracy hit on ILSVRC-12 compared to other recent reduced-precision networks. R1 was extremely positive on the paper, impressed by its readability and the quality of comparison to previous approaches (noting that results with 2-bit activations and 4-bit weights matched FP baselines). This seems very significant to me. R1 also pointed out that the technique used the same hyperparameters as the original training scheme, improving reproducibility/accessibility. R1 asked about application to MobileNets, and the authors reported some early results showing that the technique also worked with smaller network/architectures designed for low-memory hardware. R2 was less positive on the paper, with the main criticism being that the overall technical contribution of the paper was limited. They also were concerned that the paper seemed to motivate based on reducing memory footprint, but the results were focused on reducing computation. R3 liked the simplicity of the idea and comprehensiveness of the results. Like R2, they thought the paper was limited novelty. In their response to R3, the authors defended the novelty of the paper. I tend to side with the authors that very few papers target quantization at no accuracy loss. Moreover, the paper targets training, which also receives much less attention in the model compression / reduced precision literature. Is the architecture really novel? No. But does the experimental work investigate an important tradeoff? Yes.